# Defect Inspection Using Modified YoloV4 on a Stitched Image of a Spinning Tool

**DOI:** 10.3390/s23094476

**Published:** 2023-05-04

**Authors:** Bor-Haur Lin, Ju-Chin Chen, Jenn-Jier James Lien

**Affiliations:** 1Department of Computer Science and Information Engineering, National Cheng Kung University, Tainan 701, Taiwan; 2Department of Computer Science and Information Engineering, National Kaohsiung University of Science and Technology, Kaohsiung 807, Taiwan

**Keywords:** random sample consensus (RANSAC), tool rotating detection, scale-invariant feature transform (SIFT), YOLOv4

## Abstract

In Industry 4.0, automation is a critical requirement for mechanical production. This study proposes a computer vision-based method to capture images of rotating tools and detect defects without the need to stop the machine in question. The study uses frontal lighting to capture images of the rotating tools and employs scale-invariant feature transform (SIFT) to identify features of the tool images. Random sample consensus (RANSAC) is then used to obtain homography information, allowing us to stitch the images together. The modified YOLOv4 algorithm is then applied to the stitched image to detect any surface defects on the tool. The entire tool image is divided into multiple patch images, and each patch image is detected separately. The results show that the modified YOLOv4 algorithm has a recall rate of 98.7% and a precision rate of 97.3%, and the defect detection process takes approximately 7.6 s to complete for each stitched image.

## 1. Introduction

As the cost of industrial processing increases and the requirements for precision become greater, it has become a common practice to detect defects in tools before cutting. Applying tool defect detection in metal cutting machines can replace defective tools when discovered, reducing damage during the cutting process and also improving the quality of the workpiece. This, therefore, ensures that the accuracy of detection is particularly important.

In traditional measurement processes, disassembling a tool for measurement has the disadvantages of longer measurement times and potential errors, making it unable to meet the demands for efficiency. To improve this situation, this research study uses automatic optical inspection (AOI) technology. AOI is divided into three types: contact probe inspection, laser inspection, and camera inspection. This research chooses a visual-based camera inspection method. However, most camera inspection modules are static, meaning that the camera must take an image after the tool stops rotating. Some static inspection methods require the machine’s spindle to rotate at a specific angle for each shot, so if multiple images are needed, the machine must stop many times. Furthermore, if the spindle’s rotation angle cannot be controlled accurately, this inspection method becomes very difficult to carry out. Therefore, this research study proposes a dynamic inspection method that can capture tool images while the tool is rotating to avoid these problems.

This study proposes a dynamic tool defect detection method that can continuously detect defects while the tool is rotating without the need to stop the machine. Firstly, the method captures 220 images of the rotating tool through front lighting and then uses scale-invariant feature transform (SIFT) [1] technology to obtain feature points for each image and obtain a complete 360-degree stitched image. Next, a modified YOLOv4 algorithm is used, which uses atrous spatial pyramid pooling (ASPP) [2] to extract high-resolution image features that have not been convolved multiple times and integrates a detector specifically designed to detect small defects to improve the performance of detecting said defects. This modification not only preserves image details but also obtains more information. Experimental results also show that this modified architecture design is more suitable for tool defect detection and can improve recall and accuracy performance.

In simple terms, this research study proposes a method for inspecting tool defects while a machine is still running, which can save time and avoid errors caused by tool insertion and removal. The method involves two main contributions: first, capturing a 360-degree view of the cutting tool using stitching images, enabling uninterrupted operation and avoiding errors, and second, utilizing a modified version of YOLOv4 to enhance the detection of small defects.

## 2. Related Works

An online tool inspection system is an important factor in increasing the precision of manufacturing. There are three types of inspection methods, contact probe inspection, laser inspection, and camera inspection, and the results collected from each sensor have their advantages and disadvantages. The system proposed in this research adopts a camera as a sensor to detect tool size in two dimensions (2D), which is convenient compared to other methods. In contrast, contact probe inspection and laser inspection only carry out detection in one dimension (1D), so most of the research uses a camera as a sensor. Camera inspection usually works by detecting the tool’s geometry, and we will introduce some related works below.

Kurada et al. [3] suggest that machine vision plays a crucial role in the manufacturing industry and proposes a method that combines CCD and light sources to enhance the reflection of worn regions for easier detection. The work by Chen [4] mentioned that they built a simple tool geometry inspection machine equipped with a CCD camera to extract the shape of the tool, and they also used an encoder and an optical ruler to measure it. It can obtain the basic information of tools and then record it with efficient data management, so it can increase the efficiency of tool grinding and reduce the cost of machining. As a result, it can increase productivity, lower costs, and also maintain the quality of tools efficiently. Lin [5] proposed a vision-based tool inspection system equipped with a five-axis tool grinder and detachable vision inspection mechanism, and it can also extract the image online and then make an analysis of the image’s geometry. The controller of the grinder can decide the vision inspection position in the system coordinates and has user-friendly functions such as measuring and focusing automatically. It also uses subpixels to calculate digital images to increase the resolution of the inspection and then adopts Hough transform to find edge points to increase precision. As a result, it can measure the tool’s outside diameter, radius of an arc, helix angle, relief angle, and distance. Hsiao [6] used a CCD and an autofocus lens, and the mechanism has three degrees of freedom to build a small-sized inspection system. They then applied algorithms such as the least squares method, Hough transform, and the principle of stereoscopic images to estimate the inspection accuracy of small components. There is another vision-based and automated tools inspection system presented in [7] which was built with a camera, backlight, and three-axis orientation platform, and it can inspect the tool’s outside diameter, relief angle, helix angle, and axial clearance angle. Hung [8] used a micro-drill to build a system that can inspect a tool’s length and outside diameter without high deflection and orientation. Another study [9] adopted the Taguchi method to reduce error and also used a CCD and a clamp to measure tool wear precisely. Furthermore, there is also a study related to laser inspection. For example, Huang et al. [10] adopted a laser sensor to inspect a micro-drill’s diameter automatically, and it was faster and more accurate compared to manual measurement.

In recent years, there has been an increasing amount of research on using artificial intelligence (AI) technology for detecting defects in tools and various objects. Many studies have focused on developing more accurate and efficient detection systems to improve efficiency. The following literature shows the application of AI technology in defect detection. During the cutting process, the defects produced by a tool are very small, approximately between 10 μm and 100 μm. Direct detection using machine vision has the advantages of low cost, high efficiency, and sufficient accuracy. To address this problem, a neural network model called “Dtoolnet” was proposed in the work of Xue et al. [11], which can be used to detect grinding defects in diamond tools. Similarly, Ahmed [12] also proposed and developed the “DSTEELNet” convolutional neural network (CNN) architecture to improve the detection accuracy and required time for surface defects in steel strips. In addition, to improve the generality and accuracy of computer vision algorithms, Sampath et al. [13] proposed Magna-Defect-GAN, which can generate realistic and high-resolution images to improve detection accuracy. In the manufacturing industry, automatic detection technology is widely used, but metal parts often have reflection problems. To solve this problem, Cao et al. [14] proposed a photometric stereo-based defect detection system (PSBDDS) to eliminate interference from reflections and shadows. These methods not only improve the accuracy and efficiency of detection but also effectively reduce labor costs, bringing great convenience to the maintenance and management of tools.

Detection is a common task in image processing, which refers to finding all objects of pre-defined classes in an image to determine their categories and locations. It can be divided into one-stage detection, which directly regresses the object’s position and classification by analyzing features, with well-known examples such as YOLOv3 [15], SSD [16], SqueezeDet [17], and RetinaNet [18]. Two-stage detection, on the other hand, first generates some candidate regions and then performs classification through neural networks. Well-known examples include R-CNN [19], Fast R-CNN [20], Faster R-CNN [21], and Mask R-CNN [22]. Generally speaking, one-stage detection methods are faster, but their accuracy is affected by the fact that classification and regression are both performed at once. Although two-stage detection is slower, its accuracy is higher due to the pre-defined candidate regions. Of course, with the development of one-stage detection, some have achieved the accuracy of two-stage detection and have become very fast, so we used YOLOv4 [23] in our tool defect detection study and modified it to make it more suitable for detecting tools. Introduced in 2020, YOLO-v4 is a one-stage object detection algorithm based on regression that offers high precision and real-time capabilities. It combines features from various algorithms, including YOLO-v1 [24], YOLO-v2 [25], and YOLO-v3, and is composed of three components: backbone, neck, and head. The backbone component adopts CSPDarkNet53 [26], while the neck component utilizes FPN [27] and PAN [28] techniques. The head component is identical to that of YOLOv3. This study primarily made two modifications to YOLOv4: firstly, replacing SPP [29] with ASPP to extract more information; and secondly, adding a fourth detector in the head component specifically designed to detect small defects.

This paper is divided into six parts. The first two parts are the introduction and related work, while the third part explains the overall structure of the tool defect detection system. The fourth part introduces the image stitching process and the modified YOLOv4 process separately. The fifth part presents the experimental results. Finally, the sixth part presents the conclusion and future prospects, which summarize all the methods used in this study.

## 3. Spinning Tool Inspection System

The experimental environment is shown in Figure 1. The tool inspection system can be set up inside the machine. The tool is mounted on the spindle. The inspection system is set up next to the working area, including the telecentric lens, light source, and camera. The experimental steps are as follows: (1) the tool cuts for 30 s; (2) the spindle moves to the tool inspection system for inspection for 30 s; and (3) steps 1 and 2 are repeated. The steps for tool defect inspection are as follows: first, the CNC machine moves the tool to the inspection position in the x, y, and z directions, and then the CNC machine’s spindle rotates the tool. Second, the PC sends a shooting signal to the camera, and the camera captures the image of the tool online. Third, the image is sent to the PC human–machine interface through USB3.0, and then the inspection results are sent back to the CNC machine through GigE, as shown in Figure 2 and Figure 3.

### 3.1. System Setup

In this research, the camera and lens used are a Point Grey camera (model GS3-U3-41C6M-C) and a Moritex telecentric lens (Model: MML03-HR65), respectively. Point Grey Research is a high-tech company headquartered in Vancouver, Canada. In 2016, Point Grey Research was acquired by FLIR Systems and became its subsidiary. We used the Point Grey camera (model GS3-U3-41C6M-C) because it is a global shutter type camera. The advantage of the Point Grey camera is that during exposure, all the photosensitive cells are exposed simultaneously; there is no time difference in the exposure time of all the photosensitive regions. It can capture a complete picture and uses a high-speed shutter to capture clear images while the tool is rotating at the same time.

For the lens, we chose an extremely low image distortion telecentric lens. To be able to take out the tool surface contour, we use the DOF (depth of field) of 6.2 mm of the Moritex telecentric lens (Model: MML03-HR65) and a magnification of 0.3×. After the match image resolution was 18.3 μm/pixel, the FOV (field of view) was 37.55 mm × 37.55 mm. The DOF was large enough in the capturing surface profile depth to focus clearly in order to facilitate the detection surface of the tool.

### 3.2. Spinning Tool Inspection System

Figure 4 shows the architecture of the tool defect detection system, which is divided into two parts. The first part involves image acquisition where we rotate the tool at 20 rpm, taking 220 images while it rotates 370 degrees. We exceed 360 degrees to ensure that any spindle rotation errors are accounted for in the detection of all tool surfaces. 

The second part focuses on defect detection in the stitched image. First, we use SIFT to extract feature points from the 220 images and stitch them into a single image. Then, we divide this image into smaller patch images. Finally, we use a modified YOLOv4 to detect defects in these patches. We will describe the process of obtaining the tool stitched image and the modified YOLOv4 network structure in more detail.

## 4. Tool Inspection Algorithms’ Introduction

In this section, we will describe two methods: the first is using SIFT and RANSAC [30] to generate the stitched image, and the second is using the modified YOLOv4 to detect any defects in the stitched image.

### 4.1. Tool Defect Inspection Using Modified YOLOv4 on the Stitched Image

In this section, we will introduce how to stitch 220 tool images into a stitched image. Before introducing the stitching method, we will first explain how to use the camera to obtain a clear single-tool image. In order to ensure that defects on the cutting tool can be detected during high-speed rotation, the main issue is how to use a camera to photograph the rapidly rotating cutting tool in a stationary state. First, the camera’s shutter speed needs to be fast enough. A high-speed shutter can be used to solidify fast-moving objects, while a slow shutter will blur the object. A high-speed shutter means a short exposure time, while a slow shutter means a long exposure time. The camera shutter used in this study can reach a maximum of 0.032 ms, which is extremely short for a dynamic shooting shutter speed. Due to lens equipment limitations, the aperture cannot be adjusted, but all light emitted from the light source is collected and transmitted to the camera by the telephoto lens, ensuring a very high signal-to-noise ratio. The image of the cutting tool taken at a shutter speed of 0.032 ms is also very clear. In order to calculate the relationship between the shutter and the spindle speed, assuming the tool is stationary in each frame, the calculation method is derived as shown in Formulas (1)–(4), θ = 1 (degree), t = 0.032/1000 (s), and ρ = 1/6 × 1000/0.032 = 5208 rpm. It is calculated that when the spindle rotates at 5208 rpm, the images of the tool taken by the shutter of 0.032 ms are all stationary images.
(1)$$\mathrm{Speed}\;v=\rho \;\left(\mathrm{round}/\min\right)\times 360\;\left(\mathrm{degrees}/\mathrm{round}\right)\times \frac{1}{60}\left(\min/s\right)\;=\rho \times 6\;\left(\mathrm{degrees}/s\right)$$
(2)$$\mathrm{Time}\;t\;=0.032\;\mathrm{ms}=\frac{0.032}{1000}\left(s\right)$$
(3)$$\mathrm{Distance}\left(\mathrm{Degree}\right)\;\theta =1\;\left(\mathrm{degree}\right)=v\times t=\left(\rho \times 6\right)*t$$
(4)$$\rho =\frac{\theta }{6t}$$

ρ: main shaft rotation speed (round/min). θ: number of degrees that the tool has rotated under the shutter speed (degree). t: shutter speed (s).

After obtaining 220 tool images, the next step is to stitch them into a stitched image. This method consists of four steps, the first step is to use the SIFT algorithm to detect feature points in the overlapping regions of two images; this algorithm was published by David Lowe in 1999 and refined in 2004 [31]. SIFT is a widely used and powerful feature point extraction and description method, and the feature points extracted have the characteristics of resisting changes in image size, angle, and brightness. The results are shown in Figure 5a; the small dots in the figure are the feature points found using the SIFT algorithm ($u_{i}^{k}$, $v_{i}^{k}$). $u_{i}^{k}=\left[x,\;y\right]$: position of feature point k in image i, $v_{i}^{k}\in R^{1\times 128}$. Vector of feature point k in image i.

In the next step, we use the Kd-tree [32] and K-NN [33] algorithms to find all possible correspondences between the images, as shown in Figure 5b, where all the feature points are connected. In Figure 5c, we used a filtering mechanism to remove unreasonable matches based on the distance or relative coordinates of the feature points and kept good correspondences. 

In the third step, we use the random sample consensus (RANSAC) method to find the homography information between the feature points’ positions $u_{i}^{k}{\;\mathrm{and}\;u}_{j}^{k}$ and used it to stitch the images, as shown in Figure 6.
(5)$$\left[\begin{array}{c} x\prime \\ y\prime \\ w\prime \end{array}\right]=\left[\begin{array}{c} \begin{array}{ccc} h_{11} & h_{12} & h_{13} \end{array}\\ \begin{array}{ccc} h_{21} & h_{22} & h_{23} \end{array}\\ \begin{array}{ccc} h_{31} & h_{32} & h_{33} \end{array} \end{array}\right]\left[\begin{array}{c} x\\ y\\ 1 \end{array}\right]$$
(6)$$u_{i}^{k}=H_{\mathrm{ij}}u_{j}^{k}$$

$u_{j}^{k}=\left[x,\;y\right]$: the position of feature point k in image j corresponding to image i; $H_{\mathrm{ij}}$: the homography information between $u_{i}^{k}{\;\mathrm{and}\;u}_{j}^{k}$.

Finally, in the fourth step, 220 images of the tool under normal illumination were stitched together into a complete tool unwrapped image using the multi-resolution blending method, resulting in a seamless panoramic tool image as shown in Figure 5d.

### 4.2. Tool Defect Detection Using the Modified YOLOv4

Figure 7 shows the overall architecture of the modified YOLOv4, which is divided into three parts: the backbone, neck, and head. The backbone uses CSPDarknet53 to extract local features, which are divided into five blocks named C1 to C5. Each block undergoes more convolutions.

The neck is divided into two parts: a feature pyramid network (FPN) and a path aggregation network (PAN). Its main goal is to integrate information by combining the feature maps of low-resolution and high-resolution images, as well as information that is far from and close to the original image to enrich the image’s information and improve the detection results.

The head part mainly calculates the loss function. The detection loss is added together with different weights, participating in the backpropagation process to constantly refine the model.

#### 4.2.1. Backbone (CSPDarknet53)

The backbone of our network is responsible for extracting features through convolution and providing the necessary information to the rest of the network. It is the foundation and core of the complex network, and any issues with the backbone can hinder the performance of the network. Our backbone uses CSPDarknet53, which incorporates the CSP structure, Mish activation function, and atrous spatial pyramid pooling (ASPP). The architecture of CSPDarknet53 consists of multiple CBM (convolution + batch normalization + Mish activation function) and CBL (convolution + batch normalization + leaky ReLU activation function) modules, as shown in Figure 8.

#### 4.2.2. Neck

The FPN layer passes on strong semantic features from top to bottom, as shown in Figure 9. The layer mainly merges features from the upper and lower layers to improve the target detection performance, especially for small-sized targets. The information in the backbone and neck is connected through a horizontal connection, which not only preserves more information but also allows for communication between adjacent levels of features.

The PAN effectively transmits robust localization features from the bottom layer to the top layer, as shown in Figure 10. It integrates the parameters from various detection layers and different backbone layers to maintain more low-level features. The high-resolution layer contains a substantial number of edge shapes and other features. To prevent the upper layers, such as P5, from being too far away from the input, the bottom-up path reduces the distance from the original data.

#### 4.2.3. Head and Loss Function

The head of the modified YOLOv4 consists of four detectors. The inputs are the N2, N3, N4, and N5 four-layer feature maps of the path aggregation network (Figure 11a). In the detectors, candidate box size and confidence score regression are performed, followed by backpropagation to update the entire model parameters. Higher-level detectors will detect larger targets as they have more global information in their feature maps. Each detector has pre-defined anchors, which serve as reference boxes of fixed size located on each point of the feature map to determine if there is a target present in each anchor, as shown in Figure 11b. If there is a target, the size and offset differences with the ground truth are then adjusted for subsequent fine-tuning. The anchors are defined beforehand using the K-means method on the training data, and since each detector must detect 3 different size targets, a total of 12 anchors are required.

The original YOLOv4 only had three heads for detection, with the upper-level detector (low resolution) capable of detecting large defects. However, the original information of tool defect images, including small defects, is extremely important. In order to detect small defects better, the high-resolution C1 and C2 information is directly given to Detector 2 after being processed by ASPP to obtain candidate boxes (Figure 12). Therefore, the difference between the original YOLOv4 and the modified version lies in the fact that the modified version has four heads for detection (D2, D3, D4, and D5).

The total loss of the modified YOLOv4 is composed of location loss and confidence loss, represented by Formula (7):(7)$$\mathrm{Total}\;\mathrm{Loss}\;=L_{\mathrm{BBox}}=L_{\mathrm{loc}}+{\;L}_{\mathrm{conf}}$$
Formula (8) is the positional loss, which is mainly used to calculate the difference between the pre-selected box and the ground truth box.
(8)$$L_{\mathrm{loc}}=\left(1-\mathrm{IoU}\right)+\frac{d^{2}\left(b,{\;b}^{\mathrm{gt}}\right)}{c^{2}}+\;\alpha v$$

In the calculation of IoU, b represents the Prediction BBox, while $b^{\mathrm{gt}}$ represents the Ground Truth BBox, $B\cap B^{\mathrm{gt}}$ is their intersection, and $B\cup B^{\mathrm{gt}}$ is their union. The more the two BBoxes overlap, the better the prediction effect. The squared distance between the centers $d^{2}\left(b,{\;b}^{\mathrm{gt}}\right)$ represents the Euclidean distance between the centers of the Prediction BBox and the Ground Truth BBox. $d^{2}\left(b,{\;b}^{\mathrm{gt}}\right)$ = |(x’ + $\frac{w\prime }{2}$) − (x + $\frac{w}{2}$)|^2^ + |(y’ + $\frac{h\prime }{2}$) − (x + $\frac{h}{2}$)|^2^
c represents the diagonal distance of the smallest closed bounding box that can simultaneously contain the predicted box and the true box.

Formula (9a) is for confidence loss.
(9a)$$L_{\mathrm{conf}}=1_{i,j}^{\mathrm{obj}}\{-\left[\;\hat{C}\log\left(C\right)+(1-\;\hat{C})\log\left(1-C\right)\right]\}\;\;+\lambda 1_{i,j}^{\mathrm{noobj}}\left\{-\left[\;\hat{C}\log\left(C\right)+(1-\;\hat{C})\log(1-C)\right]\right\}$$
(9b)$$1_{i,j}^{\mathrm{obj}}\left\{-\left[\;\hat{C}\log\left(C\right)+(1-\;\hat{C})\log\left(1-C\right)\right]\right\}$$
In Formula (9b), when $1_{i,j}^{\mathrm{obj}}$ = 1, it means that there is an object (foreground) in the current BBox. If there is an object in the ground truth, then $\;\hat{C}$ eqauals 1 and $1-\;\hat{C}$ eqauals 0. Therefore, the larger the actual predicted C value, the smaller the value of −log(C). If there is no object in the ground truth, then $\;\hat{C}$ equals 0 and $1-\;\hat{C}$ equals 1. Therefore, the larger the actual predicted C value, the larger the value of −log(1 − C).
(9c)$$\lambda 1_{i,j}^{\mathrm{noobj}}\left\{-\left[\hat{C}\log\left(C\right)+(1-\;\hat{C})\log\left(1-C\right)\right]\right\}$$
This is used to calculate the case where there is no object in the current BBox, and the principle is the same as (9b). λ is a weight adjustment factor. As most regions in an image consist of the background, the result may be more affected by the background. Therefore, this parameter is used to adjust the balance between the foreground and background.

## 5. Experimental Results

In the recent years of Industry 4.0 and the rapid development of industrial technology, automation is the most basic demand. The requirements of workpiece quality are increasingly high and techniques for cutting are more sophisticated. The accuracy of cutting mainly relies on tool geometry. We propose a tool defect detection based on the modified YOLOv4 before the cutting process to avoid the failure of the process and also to avoid the increase in material costs. This section will discuss the data collection, evaluation metrics, and experimental results.

### 5.1. Tool Defect Data Collection

In order to detect all angles of the tool, we generated the stitched image of the tool as shown in Figure 13. A stitched image of a tool is obtained by taking several photos of a tool at multiple different angles and then stitching them together. Since there were only a few stitched images available, a patch-based method was used to augment the data. In this research, the patch size was defined as 224 × 224 pixels. A 224 × 224 sliding window was used to slide over the stitched image to generate image patches. The stride for a defect is 32 pixels. The defect area was labeled manually beforehand (the green area in Figure 13b). A patch is labeled as a defect if it overlaps with a defect.

Each defect object also has a bounding box (the yellow box shown in Figure 13b, which is denoted as *G* = (*x**, *y**, *w**, *h**)). (*x**, *y**) stands for the x and y coordinate of the left-top corner of the bounding box. *w** is the width of the bounding box and *h**. $h^{*}$ is the height. *x**, *y**, *w**, *h** are all measured in pixels.

In summary, due to the limited number of stitched images, this study used the method of cutting patch images (Figure 13c) to increase the number of images. We have 10,417 training images, 5924 validation images, and 4808 inference images.

### 5.2. Evaluation Metrics

The experimental evaluation method involves calculating the recall, precision, and F1-score for both the original YOLOv4 and the modified YOLOv4. These metrics will be calculated according to Formulas (10)–(12).
(10)$$\mathrm{Recall}=\frac{\mathrm{TP}}{\mathrm{TP}\;+\;\mathrm{FN}}$$
(11)$$\mathrm{Precision}=\frac{\mathrm{TP}}{\mathrm{TP}\;+\;\mathrm{FP}}$$
(12)$$F1-\mathrm{Score}=\frac{2\times \left(\mathrm{Precision}\;\times \mathrm{Recall}\right)}{\left(\mathrm{Precision}+\mathrm{Recall}\right)}$$

TP represents the number of cutting tools with actual defects that were correctly detected. FN represents the number of cutting tools with actual defects that were not correctly detected. Therefore, calculating recall can determine how many defects in all ground truth have been correctly detected.

Calculating precision allows one to know how many of the detected tool defects are actually existing defects among all the detected tool defects. This is because FP represents a defect that does not actually exist but is wrongly considered to exist. The F1-score is an indicator for evaluating the performance of a model’s predictions and it is the harmonic mean of precision and recall. The F1-score takes into account both the recall and the precision of the model; therefore, it gives a better reflection of the accuracy and completeness of the model’s prediction results.

### 5.3. Tool Defect Detection Results

The experiments were carried out on a computer equipped with an Intel i7-7700k CPU, 8GB RAM, and an Nvidia GTX 2080 GPU with 8GB of memory. The experimental results will be divided into two sections. The first section will compare the recall and precision of the original YOLOv4 and the modified YOLOv4. The second section, as the original YOLOv4 was found to perform poorly when dealing with small defects, will evaluate the recall and precision of the original YOLOv4 and the modified YOLOv4 for different sizes of defects.

#### 5.3.1. Experiment Results (Defect Detection Results)

In this experiment, the performance of the original YOLOv4 and the modified YOLOv4 were compared in terms of recall and precision. As seen from Table 1, the modified YOLOv4 has a higher recall, reaching 98.7%, compared to the original YOLOv4’s 94.8% which is a 3.9% increase. Similarly, the modified YOLOv4 also has a 2% increase in precision compared to the original YOLOv4. In Section 5.3.2, it will be shown that the original YOLOv4 does not perform well in detecting small defects, but the modification to YOLOv4 included the addition of a dedicated small defect detector, leading to improved recall and precision results.

#### 5.3.2. Experiment Results (Results of Detecting Different Sizes of Defects)

In this experiment, the size of the tool defects was quantified, and the performance of the original YOLOv4 and the modified YOLOv4 in detecting defects of different sizes was compared. In the study, the images of tools captured by the camera correspond to the actual sizes in the real world, with 1 pixel equal to 0.014 mm. Therefore, as shown in Table 2 and Table 3, we divided the tool defects into three parts: less than or equal to 5000 pixels, between 5000 and 10,000 pixels, and greater than or equal to 10,000 pixels. Then, we compared the recall and precision of these three parts, and the results of the experiments before and after modification of YOLOv4 are shown in the tables.

The results of the experiment show that the modified YOLOv4 outperforms the original YOLOv4 in terms of recall when detecting small defects (less than or equal to 5000 pixels) with a recall of 98.4% compared to 81.6%. In the case of larger defects (between 5000 and 10,000 pixels and greater than 10,000 pixels), the recall performance of the original and modified YOLOv4 is similar with recall values ranging from 98.1% to 99.0%.

However, the modified YOLOv4 has a higher precision than the original YOLOv4 when detecting small defects, with a precision of 93.8% compared to 83.5%. In the case of larger defects, the original YOLOv4 has higher precision values ranging from 98.3% to 98.9%, compared to the modified YOLOv4 with precision values ranging from 94.4% to 94.7%.

Therefore, this experiment found that the modified YOLOv4 significantly outperforms the original YOLOv4 in detecting small defects, making the modified version better overall than the original YOLOv4.

## 6. Conclusions

This study proposes a dynamic tool defect detection method that can continuously detect defects while the tool is rotating without the need to stop. The method first captures the rotating tool image through front lighting and then uses SIFT technology to obtain a complete 360-degree stitched image. Next, a modified version of the YOLOv4 algorithm is used, which utilizes ASPP to extract high-resolution image features that have not been convolved multiple times, and incorporates a detector specifically designed to detect small defects to improve the detection performance for such defects. This modification not only preserves image details but also obtains more information. The experimental results demonstrate that this modified architecture design is more suitable for tool defect detection and can enhance the recall rate and accuracy results.

In future research, various types of defects will be continuously added to enhance the comprehensiveness of the model, as the current amounts of tool defect data are still limited. Additionally, the time required for defect detection from image capture currently takes about 13 to 15 s. For detecting defects on a single stitching image, modified YOLOv4 takes 7.6 s, while the original YOLOv4 only takes 5 s. This indicates that there is still potential for further improvement in detection time.

## Figures and Tables

**Figure 1 sensors-23-04476-f001:**
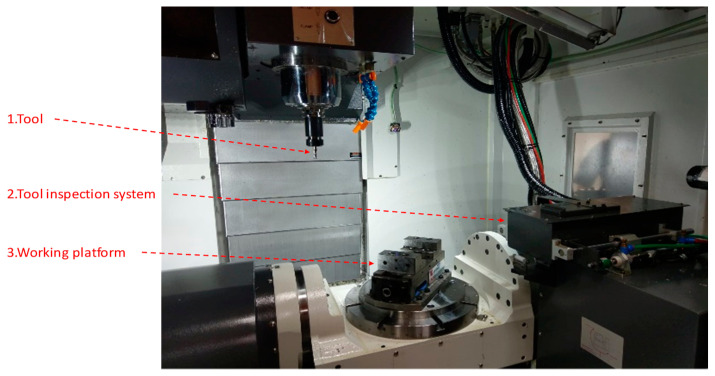
Experimental environment.

**Figure 2 sensors-23-04476-f002:**
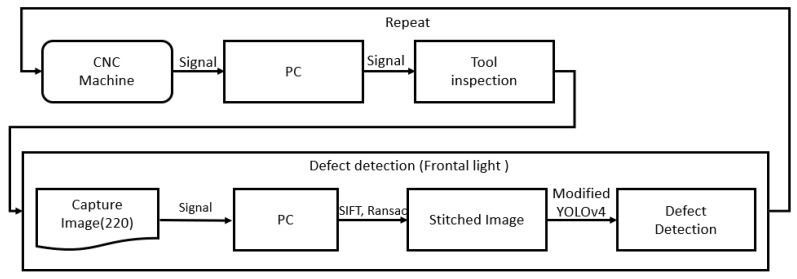
Flow chart of tool inspection.

**Figure 3 sensors-23-04476-f003:**
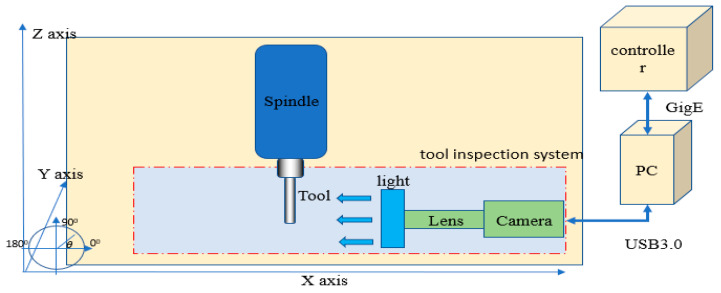
Tool inspection system.

**Figure 4 sensors-23-04476-f004:**
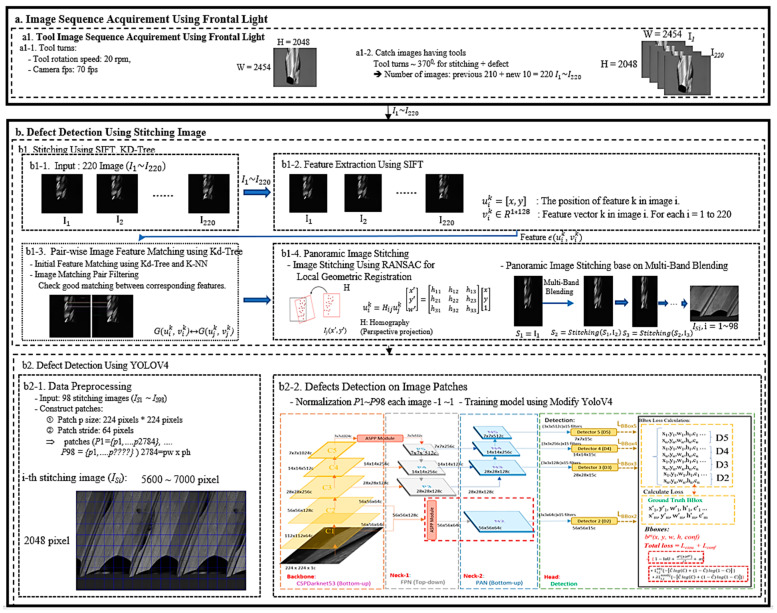
Tool defect detection system framework.

**Figure 5 sensors-23-04476-f005:**
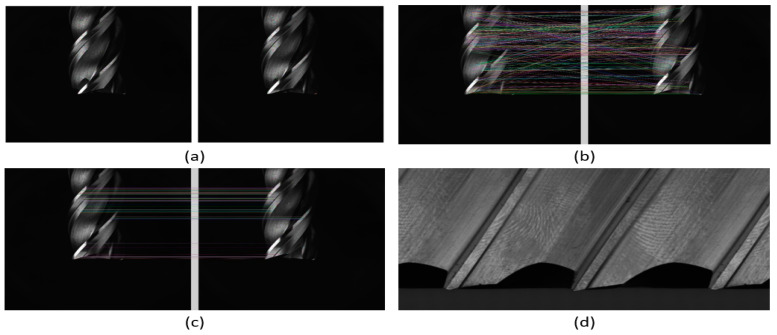
Stitching image 4-step result image. (**a**) The use of the SIFT algorithm to find local features in images. (**b**) All possible matching results between the two images. (**c**) Retainment of the best matching results. (**d**) Stitched image of the tool.

**Figure 6 sensors-23-04476-f006:**
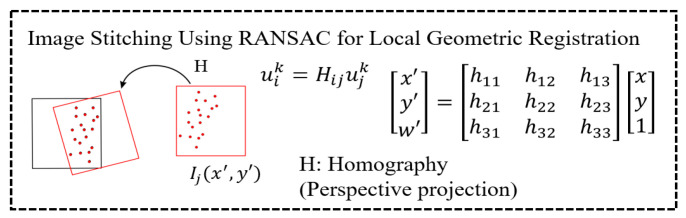
Using the RANSAC method to find homography.

**Figure 7 sensors-23-04476-f007:**
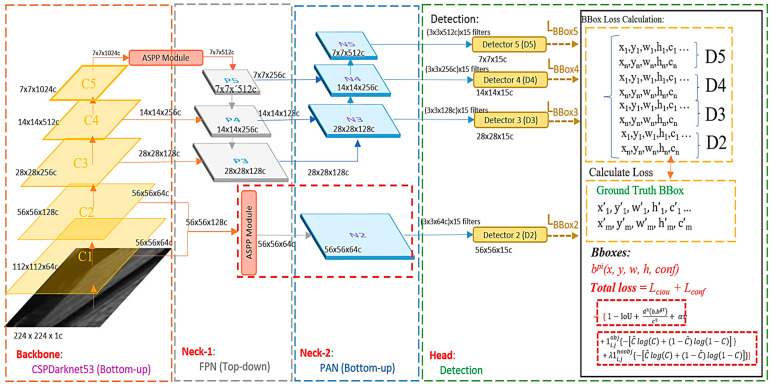
Modified YOLOv4 architecture.

**Figure 8 sensors-23-04476-f008:**
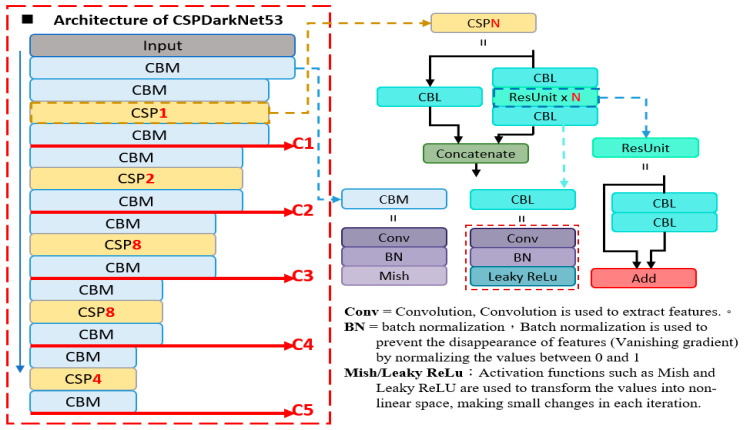
The architecture of CSPDarknet53.

**Figure 9 sensors-23-04476-f009:**
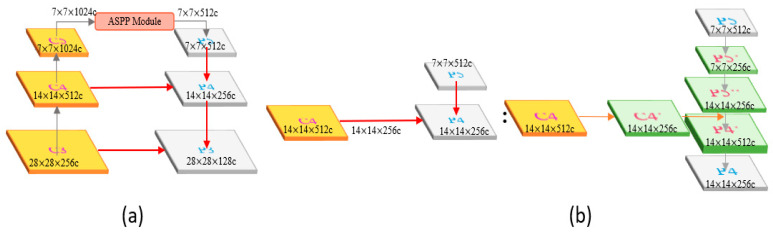
(**a**) The architecture of FPN. (**b**) The detailed architecture of FPN example.

**Figure 10 sensors-23-04476-f010:**
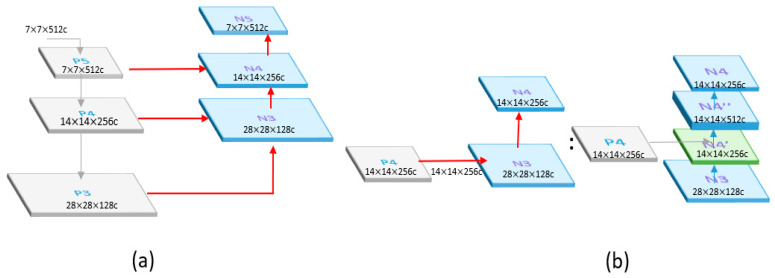
(**a**) The architecture of PAN. (**b**) The detailed architecture of PAN example.

**Figure 11 sensors-23-04476-f011:**
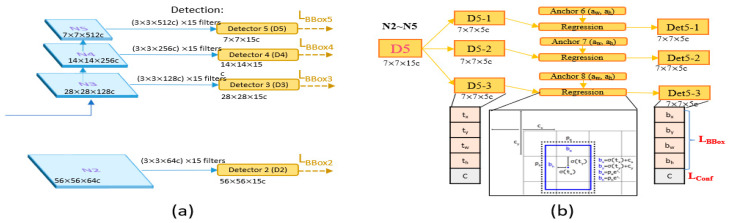
(**a**) The architecture of a detector. (**b**) The detailed architecture of Detector 5.

**Figure 12 sensors-23-04476-f012:**
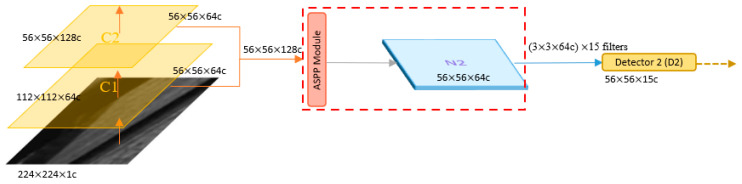
Lateral connection.

**Figure 13 sensors-23-04476-f013:**
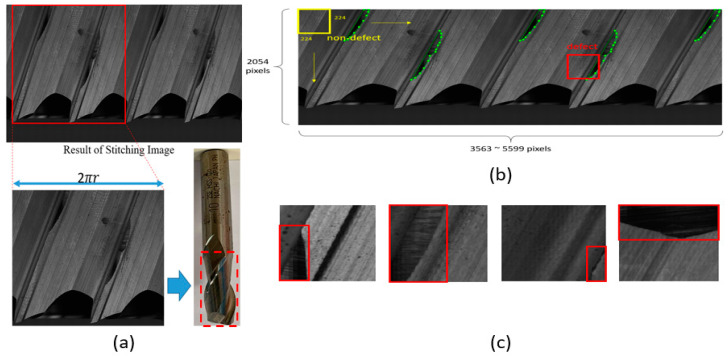
(**a**) Stitched image of a tool. (**b**) Patch generation using a 224 × 224 sliding window(The green lines indicate that the defect area was labeled manually beforehand). (**c**) The red box indicates the position of defects after cutting the image into patches.

**Table 1 sensors-23-04476-t001:** Experimental results of original YOLOv4 and the modified YOLOv4.

	Recall (%)	Precision (%)	F1-Score
YOLOv4	4588/4808 = 94.8	4558/4558 + 222 = 95.3	95.3
Modified YOLOv4	4749/4808 = 98.7	4749/4749 + 131 = 97.3	97.3

**Table 2 sensors-23-04476-t002:** YOLOv4.

	Bbox Area	Recall (%)	Precision (%)
1. Defect	≤5000 pixels	837/1025 = 81.6	837/1002 = 83.5
2. Defect	>5000 pixels and <10,000 pixels	1109/1125 = 98.5	1109/1121 = 98.9
3. Defect	≥10,000 pixels	2609/2658 = 98.1	2609/2654 = 98.3

**Table 3 sensors-23-04476-t003:** Modify YOLOv4.

	Bbox Area	Recall (%)	Precision (%)
1. Defect	≤5000 pixels	1014/1025 = 98.4	1009/1078 = 93.8
2. Defect	>5000 pixels and <10,000 pixels	1114/1125 = 99.0	1114/1176 = 94.7
3. Defect	≥10,000 pixels	2626/2658 = 98.7	2626/2779 = 94.4

## Data Availability

Data is unavailable due to the private property rights of Tongtai Machine & Tool Co., Ltd.

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
