# Peer review of "Defect Inspection Using Modified YoloV4 on a Stitched Image of a Spinning Tool"

_sensors, 2023, doi:10.3390/s23094476_

Round 1
Reviewer 1 Report
This research proposes a computer vision-based method to capture images of rotating 10 tools and detect defects without the need to stop the machine. The study uses frontal lighting to capture images of the rotating tools and employs Scale-Invariant Feature Transform to identify features of the tool images. The similarities and distinctions with the closely-related works have to be reported and thoroughly discussed. Based on the initial review, this paper does not clearly articulate the new contributions from a scientific standpoint. Here are comments for improving this paper to be publishable.
1. Problem definition: The reviewer is uncertain about the novelty of the research question. The research gap to be addressed is unclear. Please define a clear sense of the research questions resolved with the paper.
2. Abstract: The text must be carefully revised. Some sentences contain mistakes.
3. The introduction part does not have a flow or direction. It has too many different medical terminologies thrown randomly.
4. The sentences are half-constructed or incomplete, so the readers are expected to fend for themselves to understand their meaning.
5. The author must enrich the references with the latest developments. Some of the recent references can be added. The authors have not paid attention to previous research papers and concerns.
6. The deficiencies of existing studies should be discussed to highlight the characteristics and innovations of the proposed methods in the literature review section.
7. Figures are of poor quality. There are many linguistic and grammatical typos. Please carefully read through and conduct proofreading.
8. The investigation considered is a very simple and easy case. Try to involve complex problems to prove the proposed methodology.
9. With all the simplifications assumed in the investigation, the obtained results make an impression of not being an interesting and challenging exercise and, thus, unfortunately, diminish the paper's scientific merit.
The authors need to rewrite the paper or reconsider the research content before being considered for publication in this journal.
Author Response
Revirwer1:
We use the English editing system provided by MDPI on their website.
Q1: Problem definition:
The reviewer is uncertain about the novelty of the research question. The research gap to be addressed is unclear. Please define a clear sense of the research questions resolved with the paper.
A1:
Add the following paragraph to lines 53-58.
this research proposes a method for inspecting tool defects while a machine is still running, which can save time and avoid errors caused by tool insertion and removal. The method involves two main contributions: first capturing a 360-degree view of the cutting tool using stitching images, enabling uninterrupted operation and avoiding errors, second utilizing a modified version of YOLOv4 to enhance the detection of small defect.
Q2: Abstract: The text must be carefully revised. Some sentences contain mistakes.
A2:
Thank you for reminding
I have already used the editing system recommended by MDPI to polish my English.
Q3: The introduction part does not have a flow or direction. It has too many different medical terminologies thrown randomly.
A3:
1. I have included the flow chart of tool inspection in the subsequent section 3. Spinning Tool Inspection System.
2. I have already included the proprietary terms with their full names in the literature.
Q4: The sentences are half-constructed or incomplete, so the readers are expected to fend for themselves to understand their meaning.
A4:
Thank you for reminding
I have already used the editing system recommended by MDPI to polish my English.
Q5: The author must enrich the references with the latest developments. Some of the recent references can be added. The authors have not paid attention to previous research papers and concerns.
A5:
Thank you for your reminder. Recently, I have been reading more literature and incorporating it into the paper.
Q6: The deficiencies of existing studies should be discussed to highlight the characteristics and innovations of the proposed methods in the literature review section.
A6:
Thank you for your reminder. Recently, I have been reading more literature and incorporating it into the paper.
Q7: Figures are of poor quality. There are many linguistic and grammatical typos. Please carefully read through and conduct proofreading.
A7:
Thank you for your reminder. I have tried to revise some grammar and vocabulary.
Q8: The investigation considered is a very simple and easy case. Try to involve complex problems to prove the proposed methodology.
Q9: With all the simplifications assumed in the investigation, the obtained results make an impression of not being an interesting and challenging exercise and, thus, unfortunately, diminish the paper's scientific merit.
A8、9:
Due to time constraints, it is a bit difficult to conduct additional experiments. However, I have been reading more about methods for detecting tool defects and will continue to work on this aspect.

Reviewer 2 Report
The authors did an excellent job organizing and writing the scientific experiment and their results. The manufacturing industry struggles with defect detection problems; dedicated research is always extremely useful. The manuscript has been reported in a very organized manner and lucid language with proper citations. The followings are some suggestions and comments from my side to make this effort more valuable from the reader's perspective:
1. The images are taken in the optical range - Is there any chance to extend it to other spectra of electromagnetic radiation (such as x-ray)? Although I am unaware of the defect surface's material, can IR imaging be used?
2. Modified YOLOv4 has been compared with the original YOLOv4. Please add some citations/ supplements on YOLOv4 to better understand the algorithm for uninitiated readers.
3. From the result, it appears that- the modified YOLOv4 outperforms the original version when it comes to detecting more minor defects - however, the original version is still better for bigger defect detection. So now, on deployment into actual production - when we don't know the defect size (it can be smaller and bigger), how can the system handle both with great precision? Is there a chance to select versions depending on the defect size?
Author Response
Revirwer2:
We use the English editing system provided by MDPI on their website.
Q1: The images are taken in the optical range - Is there any chance to extend it to other spectra of electromagnetic radiation (such as x-ray)? Although I am unaware of the defect surface's material, can IR imaging be used?
A1:
1. Following some research, I ascertained that X-ray CT technology is employed by some individuals for identifying defects in objects. However, being unfamiliar with this area, I am uncertain of the practicality of this approach. Detecting defects in cutting tools can be arduous, given their often minute size. A high level of X-ray CT reconstruction accuracy could potentially alleviate this issue. Nonetheless, X-ray CT reconstruction does have a minor drawback, namely, it necessitates a protracted detection time and machine shutdown.
2. In my view, infrared imaging is exceptionally responsive to temperature variations. Nonetheless, shortly after processing a cutting tool, the tool's entire tip is heated, posing a challenge in distinguishing defective areas. Nevertheless, relevant literature proposes that as the size of a cutting tool defect expands, the cutting resistance also increases, leading to a temperature increase in the tool that can be discerned through infrared imaging.
Q2: Modified YOLOv4 has been compared with the original YOLOv4. Please add some citations/ supplements on YOLOv4 to better understand the algorithm for uninitiated readers.
A2:
It has already been included in the related work (line 130-139).
Q3: From the result, it appears that- the modified YOLOv4 outperforms the original version when it comes to detecting more minor defects - however, the original version is still better for bigger defect detection. So now, on deployment into actual production - when we don't know the defect size (it can be smaller and bigger), how can the system handle both with great precision? Is there a chance to select versions depending on the defect size?
A3:
1. In this study, we sacrificed a small portion of the performance for large defects in order to achieve better results for overall and small defects. As a result, in the final Precision results, the modified YOLOv4 model improved by 4.2% and 3.9% respectively at 10000 pixels and between 5000 and 10000 pixels, compared to the original YOLOv4 model. However, when the image size was less than or equal to 5000 pixels, the modified YOLOv4 model improved by 10.3% compared to the original YOLOv4 model. Ultimately, the modified YOLOv4 model outperformed the original YOLOv4 model. However, this is also a problem that we need to address in the future. In terms of recall, the modified YOLOv4 model performed better than the original YOLOv4 model.
2. If we encounter larger or smaller defects in a new environment, I believe that as long as the image accuracy is sufficient (0.014mm in this experiment), we only need to adjust the anchor size in the detectors.

Reviewer 3 Report
The authors propose a computer vision-based method to capture images of rotating tools and detect defects without the need to stop the machine. The idea is very interesting and complete, and the result is validated in practive in a real scenario. The paper seems very complete, with up to date references and sound solutions for sub-problems such as image stiching and defect recognition. In my opinion, the main problem with this paper is that is needs a careful english review, as many minor errors could be found and this may harm readers' understanding.
"The paper is divided into six parts. The first two parts are the introduction and related work, while the third part explains the overall structure of the tool defect detection system. The fourth part introduces the image stitching process and the modified YOLOv4 process separately. The fifth part presents the experimental results. Finally, the sixth part is the conclusion and future prospects, summarizing all the methods used in this study." -> this should appear in the end of the introduction section, before section 2.
One important point that is not clear in the paper is about the computational performance of the modified yolov4. It shows more accurate results in comparison to the original implementation, but how fast is it in comparison to the original one?
More general comments and minor errors are listed as follows.
"The system proposes" -> "The system proposed"
"dimensions (2D)" -> "dimensions (2D),"
"convenient compare to other method. " -> "convenient compared to other methods. "
"Paper [1] suggests" -> "Kurada et al. [1] suggest"
"machine that equipped" -> "machine equipped"
"productivity lower" -> "productivity, lower"
"In [2]" -> "The work from Chen [2]"
"In paper [3]" -> "Lin [3]"
"system which equipped" -> "system equipped"
"in the system coordinate" -> "in system coordinates"
"a user-friendly" -> "user-friendly"
"In paper [4]" -> "Hsiao [4]"
"In paper [6]" -> "Hung [6]"
"The other" -> "Another"
", in paper [8]" -> "Huang et al. [8]"
"measure manually." -> "manual measurement."
"literature [9]," -> "the work of Xue et al. [9],"
"literature [10]" -> "Ahmed [10]"
"literature [11]" -> "Sampath et al. [11]"
"literature [12]" -> "Cao et al. [12]"
What is HMI?
"cells exposed" -> "cells are exposed"
"and using high-speed" -> "and use high-speed"
"Spinning tool inspection system of the Framework" -> please rewrite
"Use RANSAC" -> "Using RANSAC"
": The position of feature point k in image i, ??? ∈ ?1×128 : The 128-dimensional unit vector describing feature point k in image i." -> please rewrite
"The Figure 7" -> "Figure 7"
Please fix the "red underscore" in Figure 8 (ReLU word).
"the figure 9 The layer" -> "Figure 9. The layer"
"The detail architecture" -> "The detailed architecture"
"The detail architecture" -> "The detailed architecture"
" by the formula (7)." -> " by formula (7)."
", In the" -> ". In the"
"their union," -> "their union."
"The formula (9)" -> "Formula (9)"
"then stitched" -> "then stitching"
"Inference" -> "inference"
"Table1: " -> "Table 1: "
Table 1 is not necessary, since all of its information is given in the text.
"This experiment evaluation method is to calculate the Recall, Precision, and F1-Score [32]of both the original YOLOv4 and the improved YOLOv4 respectively, as shown in formulas (10~12)." -> please rewrite
"Conclusion." -> "Conclusion"
Author Response
Revirwer3:
We use the English editing system provided by MDPI on their website.
Q1: "The paper is divided into six parts. The first two parts are the introduction and related work, while the third part explains the overall structure of the tool defect detection system. The fourth part introduces the image stitching process and the modified YOLOv4 process separately. The fifth part presents the experimental results. Finally, the sixth part is the conclusion and future prospects, summarizing all the methods used in this study." -> this should appear in the end of the introduction section, before section 2.4
A1:
I have already added all the methods used in this study to the end of the introduction section from line42-58
Q2: One important point that is not clear in the paper is about the computational performance of the modified yolov4. It shows more accurate results in comparison to the original implementation, but how fast is it in comparison to the original one?
A2:
It has been included in the conclusion and future work that during inference, using a single stitched image as a basis, modified YOLOv4 takes 7.6 seconds while the original YOLOv4 takes 5 seconds. The reason for the slower speed of modified YOLOv4 is the addition of detector2, which makes it slower than the original YOLOv4.
